# Childhood maltreatment and biomarkers for cardiometabolic disease in mid-adulthood in a prospective British birth cohort: associations and potential explanations

Leah Li, Snehal M Pinto Pereira, Christine Power

Population, Policy and Practice Programme, University College London Great Ormond Street Institute of Child Health, London, UK

**Correspondence to**
Dr Leah Li; leah.li@ucl.ac.uk

## ABSTRACT

**Objectives** Research on associations between childhood maltreatment and adult cardiometabolic disease risk is sparse. We aimed to investigate associations between different forms of child maltreatment and mid-adult cardiometabolic markers and whether potential intermediaries could account for the associations observed.

**Setting** 1958 British birth cohort.

**Participants** Approximately 9000 cohort members with data on cardiometabolic markers.

**Outcomes** Adult (45y) cardiometabolic markers (blood pressure, lipids and glycated haemoglobin [$HbA_{1c}$]).

**Results** Seventeen per cent of participants were identified as neglected; 6.1%, 1.6% and 10.0% were identified as experiencing physical, sexual and psychological abuse, respectively. Childhood neglect and physical abuse were associated with high body mass index (BMI) and large waist circumference when adjusting for early-life covariates. For neglect, the adjusted odds ratio (AOR) was 1.16 (95% CI: 1.02 to 1.32) and 1.15 (1.02 to 1.30) for general and central obesity, respectively, and for physical abuse, the respective AOR was 1.36 (1.13 to 1.64) and 1.38 (1.16 to 1.65). Neglect was also associated with raised triglycerides by 3.9 (0.3 to 7.5)% and $HbA_{1c}$ by 1.2 (0.4 to 2.0)%, and among females, lower high-density lipoprotein cholesterol (HDL-c) by 0.05 (0.01 to 0.08) mmol/L after adjustment. For physical abuse, the AOR was 1.25 (1.00 to 1.56) for high low-density lipoprotein cholesterol, $HbA_{1c}$ was raised by 2.5 (0.7 to 4.3)% (in males) and HDL-c was lower by 0.06 (0.01 to 0.12)mmol/L (in females). Associations for sexual abuse were similar to those for physical abuse but 95% CIs were wide. For psychological abuse, the AOR for elevated triglycerides was 1.21 (1.02 to 1.44) and HDL-c was lower by 0.04 (0.01 to 0.07)mmol/L. Maltreatments were not associated with raised blood pressure. In analyses of potential intermediary factors, several associations attenuated after adjustment for adult lifestyles (mainly smoking and alcohol consumption rather than physical activity) and child-to-adult BMI.

**Conclusions** Childhood maltreatments, particularly neglect and physical abuse, were associated with greater adiposity and poorer lipid and $HbA_{1c}$ profiles decades later in adulthood. Associations were modest but independent

## Strengths and limitations of this study

► Data were from a population-based cohort followed-up over five decades and included information on a range of adult cardiometabolic measures, childhood maltreatment (neglect and abuse), childhood covariates and potential adult intermediary factors.

► Sample reductions due to missing information were addressed using multiple imputation.

► Study power for sexual abuse may be inadequate for detecting associations due to low prevalence.

of early-life factors linked to these outcomes. Findings implicate adult lifestyles as an important intermediary between child maltreatment and outcomes.

## BACKGROUND

There is now evidence showing links between early-life adversities and later cardiovascular disease (CVD) incidence and mortality,[1–3] including some with measures of child maltreatment (neglect and abuse).[4–8] In the USA, adverse childhood experiences (ACE) study reported child neglect or abuse was found to be associated with increased risk of self-reported ischaemic heart disease (OR=1.3 to 1.7 for specific forms of maltreatment); the highest risk (OR=3.6, 95% CI: 2.4 to 5.3) was found for multiple ACEs (≥7 vs 0)[5] as also seen in a more recent US study.[6] Among 33–45-year-olds in the Coronary Artery Risk Development in Young Adults Study, a one-unit increase in risky family score (range 0–21 from self-report items) was associated with 1.0% higher 10-year CHD risk (using the Framingham algorithm).[7] Mortality follow-up studies have shown self-reported child abuse to predict mortality from all causes in the USA,[9] while those with severe (vs no) physical

abuse had increased risk for CVD events (HR=1.46 [1.11 to 1.92]) in the Nurses' Health Study.[8] Accordingly, a scientific statement from the American Heart Association identified severe childhood adversities, such as physical and sexual abuse, as emerging independent risk factors for incident ischaemic heart disease among women.[10]

However, much remains unknown about child maltreatment and CVD risk associations, including whether these are specific to different forms of maltreatment, whether associations are independent of other early-life factors and what the likely explanations or pathways are. Relevant to the latter is the growing evidence of child maltreatment and CVD risk factors, which provides important clues on possible intermediary pathways to CVD events. Investigation on a range of cardiometabolic biomarkers would be informative, but studies to date typically focus on blood pressure (BP)[11–14] or obesity,[15] and rarely include blood lipids[7 16 17] or glucose metabolism.[18–20] Study findings are not always consistent; for example, some[11] but not all[12] show associations of physical or sexual abuse with raised BP. Differences in study design may account for such discrepancies. In particular, it is important to account for other early-life factors, such as low birth weight[1] and socio-economic position (SEP),[2] linked to these outcomes. Moreover, the literature mostly focuses on child abuse[11 12 19 21] while neglect is often only included as a component of a combined measure[5 18 22] despite its association with increased adult body mass index (BMI),[23] a risk factor for the cardiometabolic disease. Combined maltreatment measures may be useful to demonstrate cumulative effects but potentially obscure important differences that provide insights into mechanisms. Thus, two recent meta-analyses of long-term health consequences of non-sexual maltreatment[3] or cumulative childhood adversity[4] concluded that research is required to confirm relationships between all forms of child maltreatment, including neglect, and later chronic disease.

With respect to explanations or pathways, notwithstanding a few exceptions,[21] research is scarce on factors underlying the associations between forms of child maltreatment and biomarkers for adult cardiometabolic disease. Yet potential mechanisms can be identified, such as allostatic load, whereby physiological wear and tear over the life course[24] may occur in response to child maltreatment and thence influence adult health. Several intermediary factors may be postulated. First, associations may be due to greater BMI gains[21 25] or obesity[11 26] among maltreated groups. Second, child maltreatment may influence adult SEP,[27] for example, via poorer education level,[28] which in turn affects adult health. Third, maltreatment may lead to risky behaviours (eg, smoking[29] or alcohol use[30]), which impact on disease risk. Fourth, the influence of maltreatment on mental health problems including depression[28] could link to cardiometabolic biomarkers via increased physiological response to stressors[31] or unhealthy behaviours.[3] We examined these potential intermediaries in a general population birth cohort. Our study aims were to investigate (1) whether different forms of childhood maltreatment (neglect and abuse) were associated with adult cardiometabolic measures (obesity, BP, blood lipids and glycated haemoglobin) independent of other early-life factors, such as birth weight and SEP; and (2) whether intermediary factors (life-course BMI gains, adult SEP, unhealthy behaviours and poor mental health) could account for associations.

## METHODS
The 1958 British birth cohort includes all births during one week in March 1958 (n=17 638) and 920 immigrants with the same birth week recruited to 16y. Information was collected throughout childhood (7, 11 and 16y) and adulthood (23, 33, 42, 45, 50 and 55y). At 45y, 11 971 individuals in contact with the study were invited to a home-based clinical assessment; 9377 (78%) participants who provided information were broadly representative of the surviving cohort.[32] The 45y survey included a childhood maltreatment questionnaire.

### Childhood maltreatment
Information was ascertained during childhood and at 45y (online supplementary table S1). Neglect was prospectively assessed (at ages 7 and 11y) using five questions to parents on their involvement with the child (infrequent outings and little interest in education) and to teachers (child's physical appearance undernourished, scruffy or dirty) that correspond to conventional definitions (details in online supplementary table S1). A scale (range 0–5) was derived by summing the five indicators separately for each age and a binary measure was derived as ≥2 items (7 and/or 11y). Child abuse (physical, sexual or psychological) by a parent up to 16y (yes/no) was reported in adulthood (45y) in a confidential self-complete questionnaire[27] (details in online supplementary table S1).

### Outcomes
Cardiometabolic measurements were obtained at 45y by nurses using standardised protocols. Height, weight and waist circumference were measured; BMI was derived as $kg/m^2$. BP was measured three times while seated after 5 minutes rest using an Omron 705CP automated digital oscillometric sphygmomanometer; means of three systolic BP (SBP) and diastolic BP (DBP) measurements were calculated. Non-fasting venous blood samples were collected and analysed for total and high-density lipoprotein cholesterol (HDL-c) and triglyceride concentrations using an autoanalyzer (Olympus AU640; Tokyo) with enzymatic methods. Low-density lipoprotein cholesterol (LDL-c) was calculated using Friedewald's formula[33] when triglyceride concentration was <4.5 mmol/L. Glycated haemoglobin ($HbA_{1c}$) concentrations were measured in whole citrated blood using ion-exchange HPLC. Nurses recorded information on prescribed medications: anti-hypertensive (n=429), lipid-regulating (n=166) and oral

**Table 1** Childhood maltreatment and adult cardiometabolic biomarkers (mean [SD] or %)

| | N | Males | Females |
|---|---|---|---|
| **Childhood maltreatment** | | | |
| Neglect* | 8734 | 18.9% | 16.0% |
| **Abuse** | | | |
| Physical | 9309 | 6.0% | 6.2% |
| Sexual | 9309 | 0.5% | 2.7% |
| Psychological | 9310 | 8.3% | 11.6% |
| **Number of types** | | | |
| 0 | 6371 | 73.4% | 73.5% |
| 1 | 1771 | 21.3% | 19.5% |
| 2 | 384 | 4.1% | 4.7% |
| ≥3 | 150 | 1.2% | 2.3% |
| **Cardiometabolic markers (45y)†** | | | |
| BMI (kg/m$^2$) | 9348 | 27.84 (4.37) | 27.00 (5.64) |
| General obesity (≥30 kg/m$^2$) | | 25.4% | 23.7% |
| Waist circumference (cm) | 9291 | 98.47 (11.24) | 85.59 (12.93) |
| Central obesity (≥102/88 cm)‡ | | 32.7% | 36.9% |
| **Blood pressure (mmHg)** | | | |
| SBP | 9297 | 133.33 (15.35) | 120.76 (16.01) |
| DBP | 9297 | 82.51 (10.84) | 76.07 (10.77) |
| High SBP/DBP (≥140/90) | 9297 | 34.6% | 16.3% |
| **Blood lipids (mmol/L)** | | | |
| Total cholesterol | 7824 | 6.10 (1.17) | 5.71 (1.02) |
| HDL-cholesterol | 7808 | 1.43 (0.34) | 1.69 (0.41) |
| Low (≤1.0/1.3) HDL-c‡ | 7847 | 11.8% | 20.3% |
| LDL-cholesterol | 7391 | 3.60 (0.96) | 3.30 (0.89) |
| High (>4.13) LDL-c | 7451 | 27.1% | 15.8% |
| Triglycerides | 7799 | 2.52 (1.83) | 1.60 (1.14) |
| High (>2.3) triglycerides | 7837 | 42.7% | 16.7% |
| HbA$_{1c}$ (%) | 7923 | 5.34 (0.82) | 5.20 (0.68) |
| High (>6%) HbA$_{1c}$ | 7964 | 6.1% | 4.2% |
| Metabolic syndrome | 7640 | 15.1% | 10.2% |

*≥2 indicators (7 and/or 11y); sample for individuals with cardiometabolic marker(s).
†All continuous measures of cardiometabolic markers were adjusted for medication. For binary outcomes (hypertension, dyslipidemia or T2 diabetes), those on medication were in risk groups. N differs as some with medication information but no blood sample measures were in risk group.
‡Cutoffs for males/females.

glucose-lowering/antidiabetic (type 2 [T2] diabetes) (n=111); T2 diabetes was also identified from self-reported doctor diagnosis at 42y (n=134) excluding type 1 (T1) diabetes (n=57).

Cardiometabolic biomarkers (binary variables) were derived using established cutoffs (table 1) from the literature[34 35] and our previous studies.[36] These include general obesity (BMI≥30 kg/m$^2$) and central obesity (waist circumference≥102 cm [males] and ≥88 cm [females][34]), hypertension (SBP/DBP≥140/90 mmHg or anti-hypertensive medication[37]), high triglycerides (>2.3 mmol/L[38]), low HDL-c (≤1.0 mmol/L [males] and ≤1.3 mmol/L [females]) and high LDL-c (>4.13 mmol/L[35]). High triglycerides, low HDL-c and high LDL-c groups also include individuals on lipid-regulating medications. Elevated (≥6%) HbA$_{1c}$[39] and T2 diabetes were grouped together, excluding T1 diabetes. Metabolic syndrome was defined using modified criteria of the Third Report of the National Cholesterol Education Program's Adult Treatment Panel[35] as ≥3 of central obesity, high BP, low HDL-c, high triglycerides and elevated HbA$_{1c}$ (or T2 diabetes) as defined above.

## Covariates

Covariates were selected a priori from the literature based on relationships with child maltreatment and adult cardiometabolic biomarkers, including factors affecting (measurement of) outcomes, early-life factors and potential intermediary factors. Early-life factors were all recorded prospectively at birth (birth weight for gestational age and childhood social class) or at age 7y (household crowding and housing tenure). Birth weight for gestational age was calculated as birth weight (measured) standardised within each gestational week. Social class was based on father's occupation at birth (or at 7y if missing); classified as I/II (professional/managerial), IIINM (skilled non-manual), IIIM (skilled manual) and IV/V (semi-unskilled manual, including no male head). Overcrowding (≥1.5 persons per room) and social housing (rented from the council or housing association vs owner-occupied or privately rented) were used as measures of material (dis)advantage.

Factors affecting (measurement of) outcomes at 45y include: measured room temperature (for BP), month of examination, time of day of blood collection, postal delay of blood sample and self-reported time since last meal (for lipids and $HbA_{1c}$), oral contraception (10.8%) and HRT use (7.4%) for females (for all outcomes) and self-reported family history of T2 diabetes (2.4%) (for $HbA_{1c}$). All factors affecting the measurement of the outcome were included in the analysis of metabolic syndrome.

*Intermediary factors* for associations between maltreatment and cardiometabolic markers included childhood (7y) and adult (45y) BMI, adult SEP (social class and educational qualifications), adult lifestyles (smoking, alcohol consumption and physical activity) and mental health as indexed by depressive symptoms. BMI was derived from measured heights and weights at ages 7y and 45y. Adult social class was based on the participant's occupation at 42y (or at 33y if missing). Educational qualifications attained by 33y was categorised into: degree/higher, A-level, O-level, <O-level and no qualifications. Smoking status was reported at 42y (or at 33y if missing). Alcohol consumption frequency was reported at 42y, classified as: non-drinker, ≤once/month, ≤3 time/month, once/week, 2–3 days/week and most days. Leisure-time physical activity, ascertained at 42y from questions on how often cohort members participated in a range of activities, was categorised as: active ≤3 times/month, 1 day/week, 2–3 days/week and 4–7 days/week.[40] For depressive symptoms at 42y, a 0–15 scale was derived from 15 psychological items included in the Malaise Inventory.[41]

## Analysis

Associations of each child maltreatment with continuous outcomes (measured by differences in mean levels) and binary outcomes (measured by ORs) were estimated using linear and logistic regressions, respectively, for genders combined, except where associations differed (p≤0.05 for gender*maltreatment interaction), which were analysed separately. Continuous triglycerides and

$HbA_{1c}$ measures were log-transformed to correct skewness of distributions; per cent change in mean level approximates was 100×(regression coefficient).[42] Models included corrections for effects of medication to minimise potential bias in estimates: (1) for antihypertensive medication (n=429), we added a commonly used constant of 10 mmHg (which has been suggested on the basis of evidence from clinical trials) to observed SBP and DBP[43]; (2) lipid levels (n=166) were corrected assuming that lipid-lowering drugs reduce total cholesterol by 20%, LDL-c by 35%, triglycerides by 15% and increase HDL-c by 5%, based on the average efficacy of statins, the most frequently prescribed lipid-lowering drug in this study[44] and (3) for antidiabetic medication (n=111), $HbA_{1c}$ levels were corrected assuming that medication reduced $HbA_{1c}$ levels by 1% in absolute terms.[45]

Baseline models include gender and factors affecting measurements (model 1). Models were then adjusted for early-life factors (and, for $HbA_{1c}$, family history of T2 diabetes) (model 2) and adjusted differences in mean levels and adjusted OR (AOR) were estimated. For associations that remained in model 2 (p<0.05, or borderline for sexual abuse due to low prevalence), we further adjusted for potential intermediary factors. To test the separate and combined intermediary role of these factors, we examined models that further adjusted for:[1] BMI at 7 and 45y,[2] SEP (social class and educational qualifications),[3] lifestyle factors (smoking, alcohol consumption and physical activity),[4] depressive symptoms and[5] all factors simultaneously (models 3–7). However, conventional regression models with adjustment for potential mediators could possibly induce bias, for example, via exposure–mediator interactions.[46] Hence, we conducted selected checks to ensure that associations were unaffected by such biases: (1) tests of maltreatment–intermediary factor interaction(s) that were mostly non-significant and (ii) adjustments to test mediator–outcome confounding: we examined effects on parameter estimates of additional adjustment for adult SEP, which is associated with both intermediary factors (eg, lifestyles) and cardiometabolic outcomes[44]; effects of such adjustments were negligible. As an additional check, we conducted a mediation analysis using inverse OR weighting method.[47] General patterns of mediation effects were consistent with those obtained from regression models 3–7 and conclusions remained unchanged. Hence, we present results using conventional regression with adjustment for mediators for simplicity of interpretation.

In addition to analyses of specific types, we examined the cumulative burden of maltreatment, as in the previous research[5]; we summed the number of types of maltreatment and using linear regression, examined associations with cardiometabolic measures, with and without adjustment for early-life factors.

As sensitivity analyses, we examined the effects of treatments (1) with adjustment for medications rather than using correction and (2) excluding those who were on treatment. Results were similar to those presented using correction for treatment (data not shown).

Among participants with cardiometabolic markers (n=7391–9297 depending on outcome), 93% had information on neglect and nearly all on abuse. To maximise available information, we applied multiple imputations to missing data on covariates and neglect to the sample alive to 45y (n=17 313). Imputation models include analysis variables (maltreatment, cardiometabolic markers and covariates) and predictors of non-response[32] (gender, ethnicity, class at birth and 7y reading ability). We created 25 imputed datasets assuming missing at random given other variables in the imputation models. Analysis samples were restricted to individuals with observed cardiometabolic markers and abuse measures (analysis sample, n=7391–9297). Parameters from imputed datasets were combined to obtain overall estimates using Rubin's rules.[48] Regression models and multiple imputations by chained equations were conducted using SPSS (version 24.0). Additional mediation analysis was conducted using STATA (version 15.0).

## Patient and public involvement

Patients and the public were not involved in the design of the study.

## RESULTS

Approximately 12.0% of participants reported any childhood abuse: 6.1% (physical), 1.6% (sexual) and 10.0% (psychological), 17.4% were classified as neglected; most were not maltreated, 20.4% had one and 6.1% had ≥2 types (table 1).

*Childhood neglect* was associated with several cardiometabolic biomarkers after adjusting for factors affecting measurement and early-life covariates: BMI and waist circumference were higher by 0.53 (95% CI:0.23 to 0.83) kg/m$^2$ and 1.23 (0.51 to 1.96) cm, respectively; AORs were higher for both general 1.16 (1.02 to 1.32) and central 1.15 (1.02 to 1.30) obesity and triglycerides were raised by 3.9 (0.3 to 7.5)% and HbA$_{1c}$ by 1.2 (0.4 to 2.0)%, and among females, HDL-c was lower by 0.05 (0.01 to 0.08) mmol/L (table 2). When considering potential intermediary factors, associations reduced (mostly disappeared) after further adjusting for child-to-adult BMI, adult SEP and lifestyles (mainly smoking and alcohol consumption rather than physical activity) (table 3 and online supplementary table S2).

*Physical abuse* was associated with higher BMI (by 0.72 [0.28 to 1.16] kg/m$^2$), waist circumference (by 1.29 [0.23 to 2.35] cm) and obesity (AOR=1.36 [1.13 to 1.64] and 1.38 [1.16 to 1.65] for general and central obesity, respectively); increased risk of high LDL-c (AOR=1.25 [1.00 to 1.56]), raised HbA$_{1c}$ in males (by 2.5 [0.7 to 4.3]%), and lower HDL-c in females (by 0.06 [0.01 to 0.12] mmol/L) (table 2). In relation to potential intermediary factors, associations largely attenuated when adjusting for adult lifestyles (notably smoking) and child-to-adult BMI (for HDL-c in females) except for BMI, waist circumference and obesity, which were unaffected (table 3 and online

supplementary table S2). *Sexual abuse* associations with biomarkers were similar or greater in magnitude than other maltreatments, for example, physical abuse, but CIs were wide due to few cases (table 2): for example, AOR of high LDL-c was 1.41 (0.89 to 2.23) after adjustment for early-life factors and attenuating to 1.26 (0.79 to 2.00) when adjusting for lifestyles, predominantly smoking (table 3 and online supplementary table S2). *Psychological abuse* was associated with elevated risk of high triglyceride levels (AOR=1.21 [1.02 to 1.44]) and low HDL-c (by 0.04 [0.01 to 0.07] mmol/L) (table 2). When considering potential intermediary factors, associations disappeared when adjusting for lifestyles (mostly smoking) and mental health (table 3 and online supplementary table S2).

Childhood maltreatments were not associated with raised BP or risk of metabolic syndrome, but since prevalence is low for sexual abuse, we cannot rule out elevated risks for either metabolic syndrome or T2 diabetes (table 2). Intriguingly, sexual abuse was associated with a reduced risk for hypertension (AOR=0.43 [0.24 to 0.74]); additional analysis for pulse pressure showed similar findings (results not presented). Whereas for other maltreatments, the null findings for BP, hypertension, metabolic syndrome or T2 diabetes are less likely to be due to the lack of statistical power. Finally, few associations were found between number of types of maltreatment and cardiometabolic outcomes after adjusting for early-life factors; exceptions included BMI and elevated triglyceride levels, the latter reflecting weak and mostly non-significant associations across all maltreatment types (table 4 and online supplementary table S3).

## DISCUSSION

Main findings include, first, all forms of child maltreatment were associated with at least one adverse cardiometabolic outcome (adiposity, blood lipids and/or HbA$_{1c}$) in mid-adulthood; although there was no evidence of adverse effects on BP. Importantly, associations were independent of birth weight and early-life SEP shown elsewhere to be associated with biomarkers for cardiometabolic disease.[1 2] Associations were consistent for neglect and physical abuse in relation to adiposity, while for other maltreatments and outcomes, associations were few, effects were modest, and evidence was weak for accumulative associations for number of types of maltreatment. Second, in relation to potential intermediary factors for the child maltreatment–cardiometabolic associations observed, these mostly disappeared after adjusting for adult lifestyles, suggesting that lifestyles may play a key mediating role, while adult SEP was important for the association of neglect but not for the physical abuse with adiposity.

## Methodological considerations

Study strengths include a population-based cohort followed-up over five decades, with a range of adult cardiometabolic measures, prospectively ascertained

**Table 2** Associations (mean difference or OR) between childhood maltreatment and adult cardiometabolic markers unadjusted and adjusted for early-life factors.

| | Neglect | | Physical abuse | | Sexual abuse | | Psychological abuse | |
|---|---|---|---|---|---|---|---|---|
| | Model 1 | Model 2 | Model 1 | Model 2 | Model 1 | Model 2 | Model 1 | Model 2 |
| **Mean difference (95% CI)*** | | | | | | | | |
| BMI (kg/m²) | **0.87 (0.59 to 1.15)** | **0.53 (0.23 to 0.83)** | **0.83 (0.39 to 1.27)** | **0.72 (0.28 to 1.16)** | 0.34 (−0.50 to 1.18) | 0.14 (−0.70 to 0.97) | **0.36 (0.01 to 0.71)** | 0.28 (−0.07 to 0.63) |
| Waist circumference (cm) | **1.88 (1.19 to 2.57)** | **1.23 (0.51 to 1.96)** | **1.44 (0.38 to 2.50)** | **1.29 (0.23 to 2.35)** | 0.15 (−1.89 to 2.19) | −0.16 (−2.19 to 1.88) | **0.89 (0.05 to 1.73)** | 0.74 (−0.10 to 1.58) |
| **Blood pressure** | | | | | | | | |
| SBP (mmHg) | **1.40 (0.51 to 2.29)** | 0.80 (−0.14 to 1.73) | −0.46 (−1.83 to 0.91) | −0.83 (−2.20 to 0.54) | **−2.90 (−5.56 to −0.24)†** | **−3.50 (−6.16 to −0.84)†** | −0.28 (−1.38 to 0.81) | −0.45 (−1.54 to 0.65) |
| DBP (mmHg) | **0.99 (0.38 to 1.59)** | 0.58 (−0.06 to 1.22) | 0.17 (−0.78 to 1.12) | −0.08 (−1.03 to 0.87) | **−2.02 (−3.84 to −0.20)†** | **−2.45 (−4.27 to −0.63)†** | 0.54 (−0.21 to 1.30) | 0.43 (−0.32 to 1.19) |
| **Blood lipids** | | | | | | | | |
| Total cholesterol (mmol/L) | 0.06 (−0.01 to 0.13) | 0.04 (−0.03 to 0.12) | 0.08 (−0.03 to 0.18) | 0.07 (−0.04 to 0.17) | 0.16 (−0.06 to 0.38) | 0.14 (−0.08 to 0.36) | −0.02 (−0.10 to 0.07) | −0.02 (−0.11 to 0.07) |
| HDL-c (mmol/L) | | | | | −0.07 (−0.14 to 0.01) | −0.05 (−0.12 to 0.02) | **−0.05 (−0.07 to −0.02)** | **−0.04 (−0.07 to −0.01)** |
| Males | −0.03 (−0.06 to 0) | −0.01 (−0.04 to 0.02) | 0.00 (−0.05 to 0.05) | 0.00 (−0.05 to 0.05) | | | | |
| Females | **−0.08 (−0.12 to −0.05)** | **−0.05 (−0.08 to −0.01)** | **−0.08 (−0.14 to −0.02)** | **−0.06 (−0.12 to −0.01)** | | | | |
| LDL-c | **0.06 (0.01 to 0.12)** | 0.05 (−0.02 to 0.11) | 0.07 (−0.02 to 0.17) | 0.06 (−0.03 to 0.16) | **0.19 (0 to 0.38)** | 0.17 (−0.02 to 0.36) | 0.01 (−0.07 to 0.08) | 0 (−0.08 to 0.08) |
| Triglycerides‡ (%) | **7.7 (4.3 to 11.1)** | **3.9 (0.3 to 7.5)** | 5.5 (−0.1 to 1.1) | 3.6 (−2.0 to 9.2) | 4.5 (−6.5 to 15.5) | 1.7 (−9.1 to 12.5) | 2.6 (−1.8 to 7.0) | 1.7 (−2.7 to 6.1) |
| HbA1c‡ (%) | | | | | **2.7 (0.3 to 5.1)** | **2.4 (0.0 to 4.8)** | 0.6 (−0.4 to 1.6) | 0.5 (−0.5 to 2.0) |
| Males | **1.7 (0.9 to 2.5)** | **1.2 (0.4 to 2.0)** | **2.8 (1.0 to 4.6)** | **2.5 (0.7 to 4.3)** | | | | |
| Females | | | 0.8 (−0.8 to 2.4) | 0.5 (−1.1 to 2.1) | | | | |
| **OR (95% CI) for elevated levels§** | | | | | | | | |
| General obesity | **1.32 (1.17 to 1.49)** | **1.16 (1.02 to 1.32)** | **1.42 (1.18 to 1.71)** | **1.36 (1.13 to 1.64)** | 1.20 (0.84 to 1.73) | 1.12 (0.77 to 1.61) | **1.17 (1.00 to 1.36)** | 1.13 (0.97 to 1.32) |
| Central obesity | **1.27 (1.13 to 1.43)** | **1.15 (1.02 to 1.30)** | **1.41 (1.19 to 1.68)** | **1.38 (1.16 to 1.65)** | 1.01 (0.72 to 1.42) | 0.96 (0.68 to 1.36) | 1.13 (0.98 to 1.30) | 1.11 (0.96 to 1.27) |
| Hypertension | **1.21 (1.07 to 1.37)** | 1.12 (0.98 to 1.27) | 1.00 (0.81 to 1.22) | 0.95 (0.77 to 1.17) | **0.46 (0.26 to 0.80)** | **0.43 (0.24 to 0.74)** | 1.02 (0.86 to 1.20) | 1.00 (0.85 to 1.18) |
| HDL-c | **1.36 (1.16 to 1.59)** | 1.15 (0.98 to 1.36) | **1.29 (1.01 to 1.65)** | 1.20 (0.94 to 1.54) | 1.43 (0.93 to 2.21) | 1.29 (0.83 to 2.01) | 1.12 (0.91 to 1.37) | 1.08 (0.88 to 1.32) |
| LDL-c | 1.09 (0.94 to 1.25) | 1.04 (0.89 to 1.21) | **1.27 (1.02 to 1.59)** | **1.25 (1.00 to 1.56)** | 1.46 (0.92 to 2.31) | 1.41 (0.89 to 2.23) | 1.00 (0.83 to 1.21) | 0.99 (0.82 to 1.19) |
| Triglycerides | **1.19 (1.04 to 1.36)** | 1.08 (0.93 to 1.24) | **1.26 (1.02 to 1.57)** | 1.20 (0.97 to 1.49) | 1.30 (0.83 to 2.03) | 1.20 (0.77 to 1.88) | **1.23 (1.04 to 1.47)** | **1.21 (1.02 to 1.44)** |
| HbA1c | **1.43 (1.11 to 1.84)** | 1.24 (0.95 to 1.61) | 1.38 (0.91 to 2.11) | 1.27 (0.83 to 1.94) | 1.53 (0.68 to 3.45) | 1.36 (0.60 to 3.09) | 1.20 (0.84 to 1.73) | 1.16 (0.81 to 1.67) |
| Metabolic syndrome | 1.22 (0.99 to 1.50) | 1.08 (0.89 to 1.32) | 1.20 (0.86 to 1.69) | 1.14 (0.81 to 1.60) | 1.48 (0.48 to 4.53) | 1.37 (0.45 to 4.20) | 1.18 (0.93 to 1.51) | 1.16 (0.91 to 1.47) |

Genders combined except when gender*maltreatment interaction p≤0.05. Estimates that reached significance with p<0.05 were bold-faced.

Model 1 includes gender and factors affecting measurement (for BP: measured room temperature; for lipids and HBA₁c: month of examination, time of blood collection, postal delay of blood sample and self-reported time since last meal and for females: oral contraception and HRT). For metabolic syndrome: all factors included.

Model 2: Model 1 factors, self-reported family history of diabetes (for HbA1c) and early-life factors (birth weight for gestational age, social class at birth, housing tenure and crowding at 7y).

*All cardiometabolic markers (continuous measures) were adjusted for medication. For binary outcomes (hypertension, dyslipidemia or T2 diabetes), those on medication were in risk groups.

†Included interaction term for gender (not separated by gender because of small Ns).

‡Log transformed and converted to per cent (NB: for HbA₁c and triglycerides parameters are per cent of units [%]).

§Details of risk groups in table 1.

**Table 3** Associations (mean difference or OR) for childhood maltreatment and adult cardiometabolic markers, adjusted for intermediary factors, separately and combined

| | Model 2 | Model 3 (+BMI 7and 45y) | Model 4 (+adult SEP) | Model 5 (+lifestyle factors) | Model 6 (+depressive symptoms) | Model 7 (+all factors) |
|---|---|---|---|---|---|---|
| **Mean difference (95% CI)*** | | | | | | |
| **Neglect** | | | | | | |
| BMI (kg/m²) | **0.53 (0.23 to 0.83)** | | **0.30 (0.0 to 0.61)** | **0.50 (0.20 to 0.80)** | **0.50 (0.20 to 0.80)** | **0.30 (0.0 to 0.60)** |
| Waist circumference (cm) | **1.23 (0.51 to 1.96)** | | 0.59 (−0.15 to 1.33) | **1.09 (0.37 to 1.81)** | **1.14 (0.41 to 1.86)** | 0.54 (−0.20 to 1.27) |
| HDL–c (mmol/L)(*females*) | **−0.05 (−0.08 to −0.01)** | −0.03 (−0.07 to 0.01) | −0.02 (−0.06 to 0.02) | −0.02 (−0.06 to 0.02) | **−0.04(−0.08 to 0)** | 0.0 (−0.03 to 0.03) |
| Triglycerides† (%) | **3.9 (0.3 to 7.5)** | 2.2 (−1.4 to 5.8) | 2.4 (−1.4 to 6.2) | 2.4 (−1.2 to 6.0) | 3.4 (−0.2 to 7.0) | 0.5 (−3.1 to 4.1) |
| HbA₁c (%)† | **1.2 (0.4 to 2.0)** | **1.0 (0.2 to 1.8)** | 0.7 (−0.1 to 1.5) | 0.7 (−0.1 to 1.5) | **1.1 (0.3 to 1.9)** | 0.4 (−0.4 to 1.2) |
| **Physical abuse** | | | | | | |
| BMI (kg/m²) | **0.72 (0.28 to 1.16)** | | **0.71 (0.27 to 1.14)** | **0.79 (0.36 to 1.22)** | **0.66 (0.22 to 1.10)** | **0.77 (0.34 to 1.21)** |
| Waist circumference (cm) | **1.29 (0.23 to 2.35)** | | **1.25 (0.19 to 2.30)** | **1.32 (0.27 to 2.37)** | 1.04 (−0.02 to 2.10) | **1.22(0.17 to 2.28)** |
| HDL–c (mmol/L)(*females*) | **−0.06 (−0.12 to −0.01)** | −0.04 (−0.10 to 0.01) | **−0.06 (−0.12 to −0.01)** | −0.04 (−0.09 to 0.02) | −0.05 (−0.11 to 0.001) | −0.01 (−0.06 to 0.04) |
| HbA₁c (%) (*males*)† | **2.5 (0.7 to 4.3)** | **2.1 (0.3 to 3.9)** | **2.4 (0.6 to 4.2)** | **1.9 (0.1 to 3.7)** | **2.3 (0.3 to 4.3)** | 1.4 (−0.4 to 3.2) |
| **Sexual abuse** | | | | | | |
| HbA₁c (%)† | **2.4(0.0 to 4.8)** | 2.2 (−0.2 to 4.6) | 1.8 (−0.6 to 4.2) | 1.2 (−1.2 to 3.6) | 2.0 (−0.4 to 4.4) | 0.9 (−1.5 to 3.3) |
| **Psychological abuse** | | | | | | |
| HDL–c (mmol/L) | **−0.04 (−0.07 to −0.01)** | **−0.04 (−0.07 to −0.01)** | **−0.04 (−0.07 to −0.01)** | −0.02 (−0.05 to 0.004) | −0.03 (−0.06 to 0.001) | −0.02 (−0.04 to 0.01) |
| **OR (95% CI) for elevated levels‡** | | | | | | |
| **Neglect** | | | | | | |
| General obesity | **1.16 (1.02 to 1.32)** | | 1.06 (0.93 to 1.21) | 1.13 (0.99 to 1.29) | **1.15 (1.01 to 1.30)** | 1.05 (0.92 to 1.20) |
| Central obesity | **1.15 (1.02 to 1.30)** | | 1.05 (0.92 to 1.18) | **1.13 (1.00 to 1.27)** | **1.13 (1.00 to 1.28)** | 1.04 (0.92 to 1.18) |
| **Physical abuse** | | | | | | |
| General obesity | **1.36 (1.13 to 1.64)** | | **1.36 (1.13 to 1.64)** | **1.38 (1.14 to 1.66)** | **1.33 (1.10 to 1.60)** | **1.37 (1.13 to 1.66)** |
| Central obesity | **1.38 (1.16 to 1.65)** | | **1.38 (1.16 to 1.64)** | **1.39 (1.17 to 1.66)** | **1.34 (1.12 to 1.60)** | **1.38 (1.16 to 1.65)** |
| LDL-c | **1.25 (1.00 to 1.56)** | 1.21 (0.96 to 1.51) | 1.24 (0.99 to 1.56) | 1.16 (0.93 to 1.46) | 1.21 (0.96 to 1.51) | 1.09 (0.87 to 1.37) |
| **Sexual abuse** | | | | | | |
| LDL-c | 1.41 (0.89 to 2.23) | 1.41 (0.88 to 2.24) | 1.38 (0.87 to 2.19) | 1.26 (0.79 to 2.00) | 1.34 (0.84 to 2.13) | 1.23 (0.76 to 1.98) |
| **Psychological abuse** | | | | | | |
| Triglycerides | **1.21 (1.02 to 1.44)** | 1.22 (1.02 to 1.47) | 1.21 (1.02 to 1.44) | 1.18 (0.99 to 1.40) | 1.15 (0.96 to 1.37) | 1.14 (0.95 to 1.37) |

NB: associations at p<0.05 (or borderline for sexual abuse) in Model 2, table 2 included here. Estimates that reached significance with *p*<0.05 were bold-faced.
Genders combined except when gender*maltreatment interaction *p*<0.05.
Model 1: factors, self-reported family history of diabetes (for HbA₁c) and early-life factors (see table 2 footnotes).
Model 2: Model 1 factors and measured BMI (7 and 45y).
Model 3: Model 2 factors and measured BMI (7 and 45y).
Model 4: Model 2 factors and adult social class (SEP) (own occupation at 42y [33y if missing] and qualifications by 33y).
Model 5: Model 2 factors and lifestyles at 42y (smoking, alcohol consumption and physical activity).
Model 6: Model 2 factors and depressive symptoms at 42y (0–15 Malaise Inventory items).
Model 7: all variables.
*All cardiometabolic markers (continuous measures) were adjusted for medication. For binary outcomes (hypertension, dyslipidemia or T2 diabetes), those on medication were in risk groups
†Log transformed and converted to per cent (NB: for HbA₁c and triglycerides parameters are per cent of units [%])
‡Details of risk groups in table 1.

**Table 4** Associations (mean difference or OR) between a number of maltreatments and cardiometabolic markers at 45y

| | Number of maltreatments Per increase* |
|---|---|
| Mean difference (95% CI)† | |
| BMI (kg/m²) | **0.33(0.17 to 0.49)** |
| Waist circumference (cm) | |
| Males | 0.23 (–0.33 to 0.79) |
| Females | **1.14(0.57 to 1.71)** |
| Blood pressure | |
| SBP (mmHg) | –0.10 (–0.61 to 0.41) |
| DBP (mmHg) | 0.18 (–0.17 to 0.53) |
| Blood lipids | |
| Total cholesterol (mmol/L) | 0.02 (–0.02 to 0.06) |
| HDL-c (mmol/L) | |
| Males | –0.01 (–0.03 to 0.01) |
| Females | **–0.04 (–0.06 to –0.02)** |
| LDL-c | 0.03 (–0.01 to 0.07) |
| Triglycerides‡ (%) | **2.2 (0.0 to 4.4)** |
| HbA$_{1c}$‡ (%) | **0.8 (0.4 to 1.3)** |
| OR (95% CI) for elevated levels§ | |
| General obesity | **1.13 (1.05 to 1.21)** |
| Central obesity | **1.12 (1.05 to 1.20)** |
| Hypertension | 1.01 (0.94 to 1.09) |
| HDL-c | **1.11(1.01 to 1.21)** |
| LDL-c | 1.05 (0.97 to 1.15) |
| Triglycerides | **1.10(1.01 to 1.20)** |
| HbA$_{1c}$ | 1.16 (0.99 to 1.36) |
| Metabolic syndrome | 1.10 (0.96 to 1.26) |

NB: analyses are for genders combined except where p≤0.05 for gender*maltreatment interaction where analyses are for males and females separately. Estimates that reached significance with p<0.05 were bold-faced.

See also online supplementary table S3.

*Mean difference or OR estimated from model adjusted for gender, factors affecting measurement (for BP: measured room temperature; for lipids and HBA$_{1c}$: examination month, time of blood collection, postal delay of blood sample and time since last meal and for females: oral contraception and HRT), family history of diabetes (for HbA$_{1c}$, diabetes), and early-life factors, including birth weight for gestational age, social class at birth, housing tenure and crowding at 7y.

†All cardiometabolic markers (continuous measures) were adjusted for medication. For binary outcomes (hypertension, dyslipidemia or T2 diabetes), those on medication were in risk groups.

‡Log transformed and converted to per cent (NB: for HbA$_{1c}$ the parameters are per cent of the units [%]).

§Details of risk groups in table 1.

childhood covariates including multiple indicators of family SEP and potential intermediary factors. Measurement of childhood neglect was recorded prospectively from multiple sources; items related to some dimensions (failure to meet a child's basic physical, emotional and education needs) but not all aspects (eg, inadequate

nutrition or shelter) of neglect.[49] As individual items may not imply neglectful behaviour, we used a score of ≥2 items. Such measures are associated with delayed childhood height growth,[50] cognitive ability and poorer qualifications,[28] in this population, as expected from the wider literature using different study designs and thereby supporting construct validity for our neglect measures. Childhood abuse was self-reported at 45y. All ascertainment methods for child maltreatment have limitations.[49] Parental report may be influenced by socially desirable responding and miss cases due to under-reporting, likewise only a small proportion of cases are identified by agencies. While adult report of abuse may be affected by recall bias or current emotional state,[51] retrospective report provides an accepted method in population studies[49]; in the 1958 cohort, it was blind to knowledge of issues to be investigated and data show expected associations with prospectively measured family dysfunction[52] and mental health,[28] suggesting good construct validity. Moreover, it has been suggested that retrospective reports show less-biased associations with objective adult outcomes (such as the risk factors for cardiometabolic disease examined here) than self-reported outcomes.[53] Some cardiometabolic measures were collected from non-fasted blood. Such measures may be inappropriate for clinical purposes (triglycerides levels tend to be lower in non-fasted samples[54] and vary by fasting duration and time of day[55]) but are often adequate for population studies given that fasting and non-fasting levels (eg, of triglycerides) are positively correlated[56] and because non-fasting levels have been found to be a significant risk factor for CVD.[57 58] Moreover, misclassification of lipid or triglyceride levels by our exposures is unlikely, and we adjusted for the time of last meal and blood collection in analyses for lipids and HbA$_{1c}$. As in other longitudinal studies, sample attrition had occurred over time. Although respondents in mid-adulthood were generally representative of the original cohort,[32] our previous study showed that individuals with childhood adversities (eg, neglect) were more likely than others to be lost to follow-up at 45y and thus, are under-represented in the present study.[52] Although the possibility of attrition bias cannot be ruled out, our previous work on child neglect associations with other adult outcomes suggests that its effect is likely to be negligible,[28] as also seen in further investigation of attrition bias in relation to glucose measures at 45y.[59] Sample reductions due to missing covariates and neglect measures were addressed using multiple imputations. Associations between types of maltreatment and outcomes estimated from imputed data (ie, among the sample with data on outcomes at 45y) were broadly similar to those obtained from samples with observed data (data not shown). Finally, study power for sexual abuse may be inadequate for detecting associations due to low prevalence. Regarding methodological approach to assess potential intermediary factors for maltreatment-cardiometabolic associations, we used conventional regression and confirmed the robustness

of our conclusions using additional checks (including inverse OR weighting method[47]).

## Interpretation and comparison with other studies

Our main finding of few modest associations between types of child maltreatment and biomarkers for cardiometabolic disease adds to the literature in several important respects. First, the adverse associations found for adult adiposity, blood lipids and HbA$_{1c}$ but not for BP, suggests the primary biological pathways through which some maltreatments could have deleterious effects on later cardiometabolic morbidity and mortality. These findings agree with others in their emphasis on adiposity[15] and lipids/glucose,[18–20] and while several studies report associations with BP,[11 13 17] not all do.[12 16] One possibility is that some BP findings reported to date are conflated by lack of control for other early-life factors, a short-coming that we have been able to address. Moreover, others suggest, as we do, that associations are small to modest.[7] Similar to our findings, the CARDIA study reported that 'risky' family environment (cold/unaffectionate family interactions, conflict, aggression, neglect and low nurturance) was associated with HDL-c level in women, but not with BP.[7] Specific types of maltreatment in our study, namely neglect and physical abuse, appeared to be particularly relevant to cardiometabolic outcomes; although as mentioned above associations for sexual abuse were comparable in magnitude. Comparison with existing research is hindered by the dearth of studies that distinguish neglect from other maltreatments, but our associations of neglect and physical abuse with increased HbA$_{1c}$ (potentially T2 diabetes) agree with evidence to date[18–20] and results for physical/sexual abuse are consistent with an increased CVD risk for women.[8] Unlike others,[5] we found little evidence of associations for number of types of maltreatment, a null finding that supports the suggestion of there being a few specific associations rather than cumulation across several types of maltreatment. Our findings contrast with previous research in other respects, for example, the lack of association for non-sexual abuse and metabolic syndrome (despite associations for key lipid and glucose components) contrasts with reported increased risk for physical abuse.[16 21] Such discrepancies may arise from our null findings for BP and other considerations, such as differences in design (eg, self-reported hypertension[11 13]) or in age.[13 17] Associations for maltreatment may vary by age, emerging later in life, as suggested by studies showing rapid gains in BP[14] and in BMI from early adulthood.[25]

For our second main finding, results suggest that adult lifestyle was a key proximal intermediary factor for several maltreatment–biomarker associations, which concurs with some,[5 7 8 60] though not all[11 19] previous studies. Our interpretation of this finding is that it suggests child neglect and abuse are associated with higher rates of smoking and obesity[11 25 29] that, in turn, affect cardiometabolic markers, such as lipids and HbA$_{1c}$. Concordant with our results, the US Nurses' Health Study found that adult lifestyle and medical risk factors accounted for much of the association between child abuse and adult CVD.[8] In relation to other potential explanations, we would expect adult SEP to be an important factor, as reported elsewhere,[61] especially for neglect rather than non-sexual abuse given its association with lower adult SEP in this cohort.[27] Our results confirm this expectation, as several associations for neglect (less so for abuse) attenuated after allowing for adult SEP. Of the other hypothesised intermediary factors examined, the child-to-adult BMI trajectory appeared to contribute to associations for neglect and physical abuse with adult lipid and HbA$_{1c}$ levels. This agrees with the US Nurses' Health Study in which adult BMI was found to account for most of the associations between physical/sexual abuse and adult T2 diabetes.[19] Lastly, for mental health, the suggestion in our study that this was an important pathway for psychological abuse is unsurprising. We might have expected it to impact other maltreatment–biomarker associations but found only negligible effects; although psychological processes other than depressive symptomatology could contribute.

## CONCLUSIONS AND IMPLICATIONS

Childhood maltreatments, particularly neglect and physical abuse, were associated with greater adiposity and poorer lipid and HbA$_{1c}$ profiles in mid-adulthood, suggesting further long-term health consequences for these groups. Although associations were only modest; importantly, they were independent of main early-life factors (birth weight and SEP) linked previously to these outcomes. Study findings suggest that adult lifestyles and child-to-adult BMI may be key intermediaries for many associations, with some evidence for additional factors, such as adult SEP. The role of child maltreatment for smoking and adiposity as intermediaries linked to later disease is relevant to clinicians and policymakers aiming to reduce later consequences associated with child maltreatment. However, whether these intermediary risk factors among child maltreatment groups would benefit from standard interventions (eg, for smoking cessation) in adulthood remains to be answered. Such interventions may be ineffective; for example, the propensity to smoke or gain BMI is biologically embedded from an early age and difficult to alter thereafter. This would strengthen further the need to develop strategies for preventing childhood maltreatment (eg, early home visiting or parenting programmes targeting those at risk for child abuse and neglect). Further work is needed on the role of life-course intermediary factors to better understand the pathways and mechanisms underlying maltreatments–cardiometabolic disease associations, including consideration of different as well as common pathways for specific maltreatments. These efforts are essential to effective strategies to reduce or prevent adult health consequences of child maltreatment.

**Contributors** LL conceptualised the study, carried out the analyses, drafted the manuscript, revised it critically for important intellectual content and approved the final version to be published. CP conceptualised the study, drafted the manuscript, revised it critically for important intellectual content and approved the final version to be published. SMPP revised the paper critically for important intellectual content and approved the final version to be published.

**Funding** This work was funded by the Department of Health Policy Research Programme through the Public Health Research Consortium (PHRC) and supported by the National Institute for Health Research Biomedical Research Centre at Great Ormond Street Hospital for Children NHS Foundation Trust and University College London. The views expressed in the publication are those of the authors and not necessarily those of the Department of Health. Information about the wider programme of the PHRC is available at http://phrc.lshtm.ac.uk. Data collection for participants at age 45 was funded by the Medical Research Council, grant G0000934. The authors are grateful to the Centre for Longitudinal Studies (CLS), University College London Institute of Education for the use of these data and the UK Data Service for making them available. However, neither CLS nor the UK Data Service bears any responsibility for the analysis or interpretation of these data.

**Competing interests** None declared.

**Patient consent for publication** Not required.

**Ethics approval** The 45y survey included a childhood maltreatment questionnaire, to which participants gave written informed consent. Ethical approval was given by the South-East Multi-Centre Research Ethics Committee (ref: 01/1/44).

**Provenance and peer review** Not commissioned; externally peer reviewed.

**Data sharing statement** NCDS data used in this research are available from the UK Data Archive (http://www.data-archive.ac.uk); applications for access to any data that forms part of the NCDS Biomedical Resource will require special license and should be submitted to clsfeedback@ioe.ac.uk.

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
