## [Reviewer comments · BMJ Open]

ARTICLE DETAILS

TITLE (PROVISIONAL)	Childhood maltreatment and biomarkers for cardiometabolic disease in mid-adulthood in a prospective British birth cohort: associations and potential explanations
AUTHORS	Li, Leah; Pinto Pereira, Snehal; Power, Christine

VERSION 1 – REVIEW

REVIEWER	Lu Ciyong Department of Medical Statistics and Epidemiology, School of Public Health Sun Yat-sen University, China
REVIEW RETURNED	20-Jun-2018

GENERAL COMMENTS	This manuscript titled “Childhood maltreatment and biomarkers for cardiometabolic disease in mid-adulthood in a prospective birth cohort: associations and potential explanations” aimed to “investigate associations between different forms of child maltreatment and mid-adult cardiometabolic markers and potential intermediaries for the associations.” by using data from “Approximately 9000 cohort members”. The study methodology seems to be sound, and the sample size is sufficient. However, several issues concerning the presentation and discussion of results should be solved to enhance the quality of their work. The following are the suggestions that the author needs to consider revising: Abstract: Obejectives: The purpose of the study is to examine the associations of child maltreatment and mid-adult cardiometabolic markers with adult cardiometabolic disease; however, the author described “potential intermediaries” as one of the predictive variable in their research objectives that should be clarified. Outcomes: how the author adjusted for covariates should not be described in this section. Results: Is the “OR” here were adjusted (e.g., “for physical abuse the OR of central obesity was 1.38 (95% CI: 1.16, 1.65)”) If so, describe as “adjusted odd ratios (AOR)” might be more accurate. “HbA1c was raised by 2.5 (0.7, 4.3)% (in males) and HDL-c was lower by 0.06 (0.01, 0.12) mmol/L (in females).” Is this part still the results of associations? What does “2.5 (0.7, 4.3)%” mean? Perhaps it should be clarified. “Associations for sexual abuse were similar to those for physical abuse but 95% CIs were wide.” and “Maltreatments were not
---

	associated with raised blood pressure.” Generally speaking, only meaningful and important results are reported in the abstract. Conclusions: the author made a conclusion that “Childhood maltreatments, particularly neglect and physical abuse, were associated with greater adiposity and poorer lipid and HbA1c profiles decades later in adulthood.” If this associations exist, the adjusted OR of both neglect physical abuse should also reported in the results section. The last sentence “Findings implicate adult lifestyles as an important intermediary factor.” should also be explained. Background: The introduction need to be more developed. Is there any meta-analysis or systematic review regarding the relationships between childhood abuse and cardiometabolic disease? What new knowledge could this study bring to current research? Whether this study has potential clinical implications for the early intervention or prevention of cardiometabolic disease? Perhaps, it should be introduced. Analysis: The statistical analysis seems to be sound, however, the method of parameter estimation should be illustrated (e.g., how is the OR/AOR value obtained), the P-value should be capitalized and italics, what statistical software is used and its version should also be illustrated. Results: Page 10: “Approximately 12% of participants reported any childhood abuse: 6% (physical), 1.6 (sexual), and 10% (psychological) abuse”, decimal digits should be unified (e.g., change 12% to 12.0%) Page 10: “Physical abuse was associated with higher BMI (by 0.72(0.28,1.16) kg/m2)”, is this the result of linear regression? If so, the regression coefficient should be reported (e.g., $\beta=0.72$, 95% CI=0.28-1.16). Moreover, numbers were given in the results that could be read from the table, usually, this should only be done for the main results (e.g., statistically significant results). Discussion: Page 11: “Second, in relation to potential intermediary factors for the child maltreatment-cardiometabolic associations observed, these mostly disappeared after adjusting for adult lifestyles, suggesting that lifestyles may play a key mediating role”, in my opinion, “adult lifestyles” is as a covariate on the associations between child maltreatment and adult cardiometabolic disease in this study, whether “adult lifestyles” has a “key mediating role” on the associations might be obtained by using other statistical methods (e.g., structural equation model (SEM)). Page 15: “These efforts are essential to effective strategies to reduce or prevent adult health consequences of child
--	--

	maltreatment.”, a more specific discussion about the intervention strategies of the author findings for preventing adult cardiometabolic disease would be good, some strategies are mentioned but more targeted suggestions should be provided based on the results (e.g., interventions to reduce childhood abuse as to the current social environment). As a population-based cohort study, participant lost to follow-up is a common bias and inevitable which is a limitation of study design, moreover, other limitations may also exist and need to be illustrated in the discussion. I also suggest put this part (Page 13: “Interpretation and comparison with other studies”) after the summary of the research (Page 11), and add a new paragraph to discuss the limitations of the study (usually before the conclusions and implications).
--	---

REVIEWER	Kristen P Kremer Kansas State University, USA
REVIEW RETURNED	09-Jul-2018

GENERAL COMMENTS	This article offers significant implications on a timely and important topic. I found this to be an interesting and enjoyable read, and I have a few suggestions for improvement. 1) In the introduction, the authors discuss several mechanism by which childhood maltreatment may be related to adult cardio-metabolic disease. One additional pathway to consider is allostatic load - the idea that the body wears down over time due to accumulated stressors. I would simply mention this in the Introduction as a possible pathway. [Reference - McEwen, B. S. (2000). Allostasis and allostatic load: implications for neuropsychopharmacology. Neuropsychopharmacology, 22(2), 108-124.] 2) In the methods, the authors should provide further details on the indicators of child maltreatment. Provide examples of the types of questions that comprise each type. A simple paragraph was not sufficient to have a good understanding of what was being measured. 3) With regards to the covariates, I felt that further family-level indicators should have been included. At minimum, family income needed to be considered. Although the authors considered SEP through father's occupation, family income is another important component of SEP, as it is highly correlated with both child maltreatment risk and future health outcomes. I can think of several occupations that may be ranked high on prestige but low on income and vice-versa. Thus, I don't think occupation is a sufficient indicator of SEP. 4) The write-up of results was adequate, but I found the tables to be overwhelming to sift through. In particular, Table 3 was confusing and it took me awhile to understand the results. I should be able to quickly look at the tables and understand what is going on. It was hard moving from Table 2 to Table 3, as the two tables presented the results in the opposite order. On Table 2, there were separate columns for each maltreatment type, versus in Table 3 which had maltreatment type across the rows. I found Table 2 to
--

	be much easier to understand and digest, and I would suggest changing Table 3 to follow this same pattern. 5) On the tables, I would recommend providing a footnote that explains what is being bolded. I'm guessing the authors bolded results that were significant with $p < 0.05$, although it's not well-explained. I would also recommend being consistent with decimals on the tables - sometimes you rounded to one decimal and sometimes to two decimals.
--	--

REVIEWER	Indre Ceponiene Lithuanian University of Health Sciences, Lithuania
REVIEW RETURNED	10-Jul-2018

GENERAL COMMENTS	This was a prospective cohort study (1958 British birth cohort) that included approximately 9000 individuals and performed several surveys during childhood and adulthood. The hypothesis of childhood maltreatment association with cardiovascular disease is important and topical. The manuscript is written in good English. However, I have a few comments/questions. The introduction is too long. The Methods section raises several questions. Use of multiple imputation may distort the data, especially when it was used to impute the data for >17K subjects when the real sample size is around 9K. How many patients were on lipid lowering, antihypertensive and glucose lowering medications? It is not correct to adjust lipids, BP and HbA1c according to mean expected effects of medications for respective conditions. These effects may differ dramatically individually due to genetic factors, absorption, drug interactions, etc. Instead, use of medications should be treated as confounding variables and adjusted accordingly in the models. In Table 1, the authors use a LDL cut-off of 4.13 mmol/l, 2.3 mmol/l for TG and 6 percent for HbA1c. Why were these cut-offs chosen? Was the population free of known CVD? Were there any exclusion criteria? Discussion is hard to follow with little insights into the pathogenetic differences (e.g. different results for WC for men and women). Why is there a general trend towards lower BMI in all abuse categories and all models, but abuse is associated with higher rates of obesity. Is the data correct? For Table 1, why is the sample size different for continuous and categorical data (HDL, LDL TG and HbA1c)? Terms such as 'causal chain' should be avoided as this type of study does not assess causality. Baseline characteristics of the sample should be expanded on the covariates used in the models (SES, depressive symptom scale, smoking and diabetes prevalence, known CVD, use of medications, etc.).
---

VERSION 1 – AUTHOR RESPONSE

Reviewer 1 - Lu Ciyong (Department of Medical Statistics and Epidemiology, School of Public Health, Sun Yat-sen University, China)

This manuscript titled “Childhood maltreatment and biomarkers for cardiometabolic disease in mid-adulthood in a prospective birth cohort: associations and potential explanations” aimed to “investigate associations between different forms of child maltreatment and mid-adult cardiometabolic markers and potential intermediaries for the associations.” by using data from “Approximately 9000 cohort

members". The study methodology seems to be sound, and the sample size is sufficient. However, several issues concerning the presentation and discussion of results should be solved to enhance the quality of their work. The following are the suggestions that the author needs to consider revising:

Abstract:

(1) Objectives: The purpose of the study is to examine the associations of child maltreatment and mid-adult cardiometabolic markers with adult cardiometabolic disease; however, the author described "potential intermediaries" as one of the predictive variable in their research objectives that should be clarified.

Response: Our second objective was to investigate whether the "potential intermediaries" could account for any observed associations between child maltreatment and adult cardiometabolic markers. These "potential intermediaries" (e.g. adult SEP, lifestyles and mental health) were likely to be predictive of adult cardiometabolic markers, but they were also on the pathways from child maltreatment to adult cardiometabolic markers. We have revised the objective in the abstract to make it clearer to readers.

(2) Outcomes: how the author adjusted for covariates should not be described in this section.

Response: As suggested by the reviewer, we have removed the sentence describing how we adjusted for these covariates.

(3) Results: Is the "OR" here were adjusted (e.g., "for physical abuse the OR of central obesity was 1.38 (95% CI: 1.16, 1.65)")? If so, describe as "adjusted odd ratios (AOR)" might be more accurate.

Response: We have changed 'OR' to 'AOR' for all the adjusted ORs in Results.

(4) "HbA1c was raised by 2.5 (0.7, 4.3) % (in males) and HDL-c was lower by 0.06 (0.01, 0.12) mmol/L (in females))." Is this part still the results of associations? What does "2.5 (0.7, 4.3) %" mean? Perhaps it should be clarified.

Response: These are measures of associations, depending on the distribution of the outcome. For continuous outcomes (e.g. HDL-c), the association was measured as the difference in mean HDL-c between exposed and unexposed groups (or for every unit increase in the level of the exposure). For continuous outcomes with a skewed distribution (e.g. triglycerides and HbA1c), they were log-transformed; the regression coefficient (β) was transformed to % change in mean level (approximates $100 * \beta(1)$).

"HbA1c was raised by 2.5 (0.7, 4.3) % (in males)" means that HbA1c level was 2.5% higher on average in association with child physical abuse in males. Due to the word limit for the abstract, we are unable to provide details of different measures for associations used here (i.e. absolute measures such as difference in mean levels, and relative measures such as % change or odds ratio), although this issue is described on p9 in the main text (first paragraph in Analysis).

(5) “Associations for sexual abuse were similar to those for physical abuse but 95% CIs were wide.” and “Maltreatments were not associated with raised blood pressure.” Generally speaking, only meaningful and important results are reported in the abstract.

Response: We appreciate the reviewer’s view that ‘only meaningful and important results are reported in the abstract’. However, null findings from our study are informative, as other ‘significant’ results, and both need to be reported to provide evidence on likely outcomes, and also importantly, to avoid publication bias.

(6) Conclusions: the author made a conclusion that “Childhood maltreatments, particularly neglect and physical abuse, were associated with greater adiposity and poorer lipid and HbA1c profiles decades later in adulthood.” If this associations exist, the adjusted OR of both neglect physical abuse should also reported in the results section. The last sentence “Findings implicate adult lifestyles as an important intermediary factor.” should also be explained.

Response: As suggested by the reviewer, we now report adjusted ORs and differences for neglect in the results section.

We have also revised the last sentence (abstract) to make the intermediary role clearer for readers.

Background:

(7) The introduction need to be more developed. Is there any meta-analysis or systematic review regarding the relationships between childhood abuse and cardiometabolic disease? What new knowledge could this study bring to current research? Whether this study has potential clinical implications for the early intervention or prevention of cardiometabolic disease? Perhaps, it should be introduced.

Response: We have made two main edits to the introduction. First, in addition to the systematic review and meta-analysis already quoted (2) we now include a newly published meta-analysis on cumulative childhood adversity and adult cardiometabolic disease (3). Both of these meta-analyses point to the need to understand associations for separate types of childhood adversities, which is now mentioned at the end of the second paragraph in the introduction.

Second, we have added text in response to reviewer 2 (point 16 below), regarding allostatic load as a potential mechanism linking child maltreatment and cardiometabolic outcomes, but are constrained to add further points by reviewer 3’s request that we shorten the introduction (point 21 below). However, we have added clinical implications to the discussion (point 13 below).

Analysis:

(8) The statistical analysis seems to be sound, however, the method of parameter estimation should be illustrated (e.g., how is the OR/AOR value obtained), the P-value should be capitalized and italics, what statistical software is used and its version should also be illustrated.

Response: We have edited the text (1st and 2nd Paragraphs of Analysis) to make it clearer how associations were estimated (e.g. differences in mean levels for continuous outcomes from linear regressions and ORs/AORs for binary outcomes from logistic regression). We also present P-values

with capital letters and italics, and add information on the statistical software (and its version) used to the last paragraph of the Methods.

Results:

(9) Page 10: “Approximately 12% of participants reported any childhood abuse: 6% (physical), 1.6 (sexual), and 10% (psychological) abuse”, decimal digits should be unified (e.g., change 12% to 12.0%).

Response: we have now provided all results with 1 decimal point.

(10) Page 10: “Physical abuse was associated with higher BMI (by 0.72(0.28,1.16) kg/m²)”, is this the result of linear regression? If so, the regression coefficient should be reported (e.g., $\beta=0.72$, 95% CI=0.28-1.16).

Response: We interpreted the regression coefficient for physical abuse ($\beta=0.72$) as “the difference in mean BMI between abused and non-abused individuals”, which is widely used in epidemiological research and acceptable to readers.

(11) Moreover, numbers were given in the results that could be read from the table, usually, this should only be done for the main results (e.g., statistically significant results).

Response: We report the important findings from the tables, including some estimates that were non-significant (i.e. null findings). See our response to point 5.

Discussion:

(12) Page 11: “Second, in relation to potential intermediary factors for the child maltreatment-cardiometabolic associations observed, these mostly disappeared after adjusting for adult lifestyles, suggesting that lifestyles may play a key mediating role”, in my opinion, “adult lifestyles” is as a covariate on the associations between child maltreatment and adult cardiometabolic disease in this study, whether “adult lifestyles” has a “key mediating role” on the associations might be obtained by using other statistical methods (e.g., structural equation model (SEM)).

Response: “Adult lifestyles” are covariates which are potentially on the pathway(s) between child maltreatment and adult cardiometabolic disease. To test the potential intermediary role of factors (such as adult lifestyles) in the observed associations, we used conventional regression models with adjustment for mediators (i.e. including these factors as covariates).

As mentioned by the reviewer, we also conducted mediation analysis, but using the recently developed Inverse Odds Ratio Weighting method (4) rather than SEM. This is referred to in the text as a sensitivity analysis for continuous outcomes. Patterns of mediation effects were consistent with those using conventional regression methods. As this information is given at the end of 2nd paragraph in the ‘Analysis’ section, no further additions to the manuscript have been made.

(13) Page 15: “These efforts are essential to effective strategies to reduce or prevent adult health consequences of child maltreatment.”, a more specific discussion about the intervention strategies of

the author findings for preventing adult cardiometabolic disease would be good, some strategies are mentioned but more targeted suggestions should be provided based on the results (e.g., interventions to reduce childhood abuse as to the current social environment).

Response: We have edited the concluding paragraph by adding:

'The role of child maltreatment for smoking and adiposity as intermediaries linked to later disease is relevant to clinicians and policy-makers aiming to reduce later consequences associated with child maltreatment. However,' and

"This would strengthen further the need to develop strategies for preventing childhood maltreatment (e.g. early home visiting or parenting programmes targeting those at risk for child abuse and neglect)."

(14) As a population-based cohort study, participant lost to follow-up is a common bias and inevitable which is a limitation of study design, moreover, other limitations may also exist and need to be illustrated in the discussion.

Response: We have expanded the study limitations in relation to sample attrition and missing data as follows:

'As in other longitudinal studies, sample attrition had occurred over time. Although respondents in mid-adulthood were generally representative of the original cohort, our previous study showed that individuals with childhood adversities (e.g. neglect) were more likely than others to be lost to follow-up at 45y (5) and thus are under-represented in the present study (6). Although the possibility of attrition bias cannot be ruled out, our previous work on child neglect associations with other adult outcomes suggests that its effect is likely to be negligible(7), as seen also in further investigation of attrition bias in relation to glucose measures at 45y (8). Sample reductions due to missing covariates and neglect measures were addressed using multiple imputation. Associations between types of maltreatment and outcomes estimated from imputed data (i.e. amongst the sample with data on outcomes at 45y) were broadly similar to those obtained from samples with observed data (data not shown)."

Other limitations of our study are mentioned in Methodological considerations (Discussion), including self-reported childhood abuse at 45y, non-fasted blood for some cardiometabolic measures, and inadequate study power for sexual abuse due to low prevalence.

(15) I also suggest put this part (Page 13: "Interpretation and comparison with other studies") after the summary of the research (Page 11), and add a new paragraph to discuss the limitations of the study (usually before the conclusions and implications).

Response: We carefully considered the reviewer's suggestion but decided to keep the current order as we feel it is important to outline methodological considerations (including strengths and limitations of the study) before interpreting results, making comparisons with other studies, and exploring possible explanations for these findings.

Reviewer 2: Kristen P Kremer (Kansas State University, USA)

This article offers significant implications on a timely and important topic. I found this to be an interesting and enjoyable read, and I have a few suggestions for improvement.

Response: Thank you for this positive assessment.

(16) In the introduction, the authors discuss several mechanism by which childhood maltreatment may be related to adult cardio-metabolic disease. One additional pathway to consider is allostatic load - the idea that the body wears down over time due to accumulated stressors. I would simply mention this in the Introduction as a possible pathway. [Reference - McEwen, B. S. (2000). Allostasis and allostatic load: implications for neuropsychopharmacology. *Neuropsychopharmacology*, 22(2), 108-124.]

Response: We have added this point to the last paragraph of Introduction –

“Yet potential mechanisms can be identified, such as allostatic load, whereby physiological wear-and-tear over the life course (9) may occur in response to child maltreatment and thence influence adult health.”

(17) In the methods, the authors should provide further details on the indicators of child maltreatment. Provide examples of the types of questions that comprise each type. A simple paragraph was not sufficient to have a good understanding of what was being measured.

Response: We have added further information on the indicators of child maltreatment to the 2nd paragraph of the Methods section.

Full details on questions asked, age of data collection, and ascertainment method are given in supplemental Table S1 and previous work supporting construct validity of our child maltreatment indicators (on physical, cognitive and emotional outcomes) is referred to in the methodological considerations section of the discussion.

(18) With regards to the covariates, I felt that further family-level indicators should have been included. At minimum, family income needed to be considered. Although the authors considered SEP through father's occupation, family income is another important component of SEP, as it is highly correlated with both child maltreatment risk and future health outcomes. I can think of several occupations that may be ranked high on prestige but low on income and vice-versa. Thus, I don't think occupation is a sufficient indicator of SEP.

Response: Unfortunately, information is not available for family income. Parental occupation may be an incomplete indicator of family SEP, but in addition we have included recognised indicators of material aspects of socio-economic circumstances, namely housing tenure and household crowding (10). Thus, we use multiple indicators of family SEP rather than rely on one, and all indicators have the advantage of being prospectively recorded (see methodological considerations section of the discussion).

(19) The write-up of results was adequate, but I found the tables to be overwhelming to sift through. In particular, Table 3 was confusing and it took me awhile to understand the results. I should be able to quickly look at the tables and understand what is going on. It was hard moving from Table 2 to Table 3, as the two tables presented the results in the opposite order. On Table 2, there were separate

columns for each maltreatment type, versus in Table 3 which had maltreatment type across the rows. I found Table 2 to be much easier to understand and digest, and I would suggest changing Table 3 to follow this same pattern.

Response: We agree that Table 3 is not as easy to follow as Table 2 and have carefully considered alternative formats to facilitate readability. Unfortunately, we cannot use the same format as Table 2 which includes only 2 models (columns) for each maltreatment type compared to 6 models in Table 3.

We have retained the original format of Table 3 as this has several features in common with Table 2 (similar ordering of outcomes and separation of continuous and binary outcomes) as alternative formats considered did not prove to be any better.

(20) On the tables, I would recommend providing a footnote that explains what is being bolded. I'm guessing the authors bolded results that were significant with $p < 0.05$, although it's not well-explained. I would also recommend being consistent with decimals on the tables - sometimes you rounded to one decimal and sometimes to two decimals.

Response: We have added a footnote to the tables to explain that estimates reaching significance with $P < 0.05$ were bold-faced (main tables 2-4 and supplemental tables S2-S3).

We have checked to ensure consistent decimal point(s) in all tables, i.e. one decimal point for percentages and two decimal points for differences in mean levels or odds ratios.

Reviewer 3: Indre Ceponiene (Lithuanian University of Health Sciences, Lithuania)

This was a prospective cohort study (1958 British birth cohort) that included approximately 9000 individuals and performed several surveys during childhood and adulthood. The hypothesis of childhood maltreatment association with cardiovascular disease is important and topical. The manuscript is written in good English. However, I have a few comments/questions.

(21) The introduction is too long.

Response: we have shortened the introduction by about 10%, whilst also incorporating reviewers' suggested additions (i.e. reviewer 1 point 7 and reviewer 2 point 16).

(22) The Methods section raises several questions. Use of multiple imputation may distort the data, especially when it was used to impute the data for >17K subjects when the real sample size is around 9K.

Response: Our analysis sample includes individuals with observed biomarkers (outcomes) and abuse measures at 45y. We imputed prospectively collected child neglect and covariates, but not abuse and outcomes. Using the full >17000 sample maximises the available information on which to base imputations. Restricting the imputation to the ~9K sample with data at 45y ignores valuable information and potentially distorts the imputation. However, we base analyses on the ~9k with observed outcome data, as now clarified in the last paragraph of the analysis section. We have also added further text to the discussion of missing data and attrition (see response to reviewer 1 point 14). Therefore the possibility of bias due to sample attrition cannot be ruled out. We have included the issue of attrition in limitations of the study (Discussion).

(23) How many patients were on lipid lowering, antihypertensive and glucose lowering medications? It is not correct to adjust lipids, BP and HbA1c according to mean expected effects of medications for respective conditions. These effects may differ dramatically individually due to genetic factors, absorption, drug interactions, etc. Instead, use of medications should be treated as confounding variables and adjusted accordingly in the models.

Response: Our study cohort comprises a relatively healthy population and only a small proportion of individuals were on treatments at 45y. We have added information for N on medications to the first paragraph of the analysis section: 429 on antihypertensive medications, 166 on lipid regulating medications, and 111 on glucose lowering medications. We made the same correction to readings of lipids, BP and HbA1c for all individuals on medications based on the evidence from clinical trials or average efficacy of commonly used drugs (e.g. statins).

We agree with the reviewer that it is not ideal to correct measurements according to mean expected effects of medications as effects may differ due to individual differences. We conducted sensitivity analyses to examine associations (i) with adjustment for medications (i.e. as a covariate in the model) rather than using correction and (ii) excluding those who were on treatments. We have edited the methods section accordingly by adding these sensitivity analyses and report that results were similar to those presented using correction for treatment (data not shown).

(24) In Table 1, the authors use a LDL cut-off of 4.13 mmol/l, 2.3 mmol/l for TG and 6 percent for HbA1c. Why were these cut-offs chosen? Was the population free of known CVD? Were there any exclusion criteria?

Response: According to the NCEP guidelines (11), an LDL-C level can be evaluated as: <130 mg/dL (acceptable for individuals with no other risk factors), 130-159 mg/dL (borderline high), 160-189 mg/dL (high), and ≥ 190 mg/dL (very high). We defined the high LDL-C group using a cut-off of 160 mg/dL (4.13mmol/L). Lower HDL-C level is associated with increased risk of heart disease, with <40 mg/dL for men and <50 mg/dL for women considered to be a major risk factor; 41-59 mg/dL is considered to be borderline low and ≥ 60 mg/dL as optimal. We used a cut-off of 40 mg/dL (1.0 mmol/L) for men and 50 mg/dL (1.3 mmol/L) for women to define the low HDL-C group.

Triglycerides (12) can be evaluated as: <150 mg/dL (normal), 150-200 mg/dL (borderline), and >200 mg/dL (high). We used a cut-off of 200 mg/dL (2.3mmol/L) to define the high triglycerides group.

For HbA1c, we have replaced the reference (13) with (14) (a report from an International Expert Committee on the role of A1C Assay in the Diagnosis of Diabetes). It recommends an HbA1c cut-off of $\geq 6.5\%$ for the diagnosis of diabetes and $\geq 6.0\%$ (to 6.5%) for the identification of high risk of diabetes. We used the cut-off of $\geq 6.0\%$ to identify those at high risk of diabetes.

All references for cut-offs are provided in the manuscript (paragraph 4 of methods section).

All individuals identified using these cut-offs or who were on medications for cholesterol or diabetes were classified as 'at risk' groups.

(25) Discussion is hard to follow with little insights into the pathogenetic differences (e.g. different results for WC for men and women).

Response: The majority of associations found in our study were similar for both sexes. For the few exceptions, associations were in the same direction in general, albeit slightly stronger for one sex.

Some have suggested that females may be more vulnerable to childhood adversity (3) but the evidence is not well-developed and may depend on the predominant intermediaries (e.g. obesity, smoking). Hence, we considered that the biological mechanisms of the few gender differences in associations are beyond the scope of this paper and therefore, we did not include in our discussion.

(26) Why is there a general trend towards lower BMI in all abuse categories and all models, but abuse is associated with higher rates of obesity. Is the data correct?

Response: Our results are consistent in showing a general trend towards higher BMI as well as higher rate of obesity in all abuse categories and all models. For example, in Tables 2 and 3, the differences in mean BMI were positive and the ORs for obesity were greater than one in all abuse categories.

For Table 1, why is the sample size different for continuous and categorical data (HDL, LDL TG and HbA1c)?

Response: The sample size differed slightly for continuous and categorical data (HDL, LDL TG and HbA1c). This is due to a small number of individuals who did not have measurements from the blood sample, but who provided their medications for classification by the nurse during the interview, and who were classified in the risk group. We have added a footnote to table 1 to clarify the discrepancy.

Terms such as 'causal chain' should be avoided as this type of study does not assess causality. Baseline characteristics of the sample should be expanded on the covariates used in the models (SES, depressive symptom scale, smoking and diabetes prevalence, known CVD, use of medications, etc.).

Response: We have removed the reference to 'causal chain' from the Discussion to avoid confusion.

Also, we now expand on the description of covariates used in the models in the methods section to clarify whether the characteristics were measured or self-reported and age of ascertainment to indicate prospective assessment. For example, we now mention that birthweight and BMI at 7y were measured prospectively.

In some instances (i.e. for physical activity and depressive symptoms) we have added references to provide further background details for potential intermediary factors.

For all tables, we have checked footnotes to ensure that covariates used in models are comprehensive.

Reference List

- (1) Cole TJ. Sympercents: symmetric percentage differences on the 100 log(e) scale simplify the presentation of log transformed data. *Stat Med* 2000;19(22):3109-25.
- (2) Norman RE, Byambaa M, De R, Butchart A, Scott J, Vos T. The long-term health consequences of child physical abuse, emotional abuse, and neglect: a systematic review and meta-analysis. *PLoS Med* 2012;9(11):e1001349.
- (3) Jakubowski KP, Cundiff JM, Matthews KA. Cumulative childhood adversity and adult cardiometabolic disease: A meta-analysis. *Health Psychol* 2018;37(8):701-15.
- (4) Nguyen QC, Osypuk TL, Schmidt NM, Glymour MM, Tchetgen Tchetgen EJ. Practical guidance for conducting mediation analysis with multiple mediators using inverse odds ratio weighting. *Am J Epidemiol* 2015;181(5):349-56.

- (5) Atherton K, Fuller E, Shepherd P, Strachan DP, Power C. Loss and representativeness in a biomedical survey at age 45 years: 1958 British birth cohort. *J Epidemiol Community Health* 2008;62(3):216-23.
- (6) Denholm R, Power C, Thomas C, Li L. Child maltreatment and household dysfunction in a British birth cohort. *Child Abuse Review* 2013;22:340-53.
- (7) Geoffroy MC, Pinto PS, Li L, Power C. Child Neglect and Maltreatment and Childhood-to-Adulthood Cognition and Mental Health in a Prospective Birth Cohort. *J Am Acad Child Adolesc Psychiatry* 2016;55(1):33-40.
- (8) Seaman SR, White IR, Copas AJ, Li L. Combining multiple imputation and inverse-probability weighting. *Biometrics* 2012;68(1):129-37.
- (9) McEwen BS. Allostasis and allostatic load: implications for neuropsychopharmacology. *Neuropsychopharmacology* 2000;22(2):108-24.
- (10) Galobardes B, Shaw M, Lawlor DA, Lynch JW, Davey SG. Indicators of socioeconomic position (part 2). *J Epidemiol Community Health* 2006;60(2):95-101.
- (11) Executive Summary of The Third Report of The National Cholesterol Education Program (NCEP) Expert Panel on Detection, Evaluation, And Treatment of High Blood Cholesterol In Adults (Adult Treatment Panel III). *JAMA* 2001;285(19):2486-97.
- (12) Miller M, Stone NJ, Ballantyne C, Bittner V, Criqui MH, Ginsberg HN, et al. Triglycerides and cardiovascular disease: a scientific statement from the American Heart Association. *Circulation* 2011;123(20):2292-333.
- (13) Thomas C, Hypponen E, Power C. Prenatal exposures and glucose metabolism in adulthood: are effects mediated through birth weight and adiposity? *Diabetes Care* 2007;30(4):918-24.
- (14) International Expert Committee report on the role of the A1C assay in the diagnosis of diabetes. *Diabetes Care* 2009;32(7):1327-34.

VERSION 2 – REVIEW

REVIEWER	Kristen P. Kremer Kansas State University, USA
REVIEW RETURNED	14-Nov-2018

GENERAL COMMENTS	No further comments.
----------------------

REVIEWER	Indre Ceponiene Lithuanian University of Health Sciences, Lithuania
REVIEW RETURNED	16-Nov-2018

GENERAL COMMENTS	All my comments have been adequately addressed. I have no further comments.
---